# Cohabitation in sub-Saharan Africa: Does women empowerment matter? Insights from the demographic and health survey

Castro Ayebeng[1,2]*, Joshua Okyere[2,3], Nancy Arthur[4], Kwamena Sekyi Dickson[2]

1 School of Demography, Australian National University, Canberra, ACT, Australia, 2 Department of Population and Health, University of Cape Coast, Cape Coast, Ghana, 3 School of Human and Health Sciences, University of Huddersfield, Queensgate, Huddersfield, England, United Kingdom, 4 Department of Research and Advocacy, Challenging Heights, Winneba, Ghana

* castro.ayebeng@stu.ucc.edu.gh

## Abstract

### Background

Cohabitation is increasingly challenging traditional family structures in sub-Saharan Africa (SSA), marking a significant cultural shift in the region. This rise in non-traditional unions prompts an exploration into the underlying factors, particularly the influence of women's empowerment. Despite the growing prevalence of cohabitation, there remains a notable gap in research examining its connection to women empowerment. This study aims to bridge that gap by investigating the intricate relationship between key indicators of women's empowerment and cohabitation in SSA, offering fresh insights into how changing gender dynamics may be reshaping intimate partnerships in the region.

### Methods

The study is a secondary data analysis of the Demographic and Health Survey (DHS) data of 13 SSA countries. A sample of 124,183 women between the ages of 15 and 49 having information on the outcome of interest were included in the analysis. Descriptive and inferential analyses, including proportions, Pearson's Chi-squared test, and multivariable logistic regression models, were employed to examine the association between women's empowerment indicators and cohabitation. The final regression model is presented in adjusted odds ratios (aOR) with 95% confidence intervals (CI).

### Results

We found that cohabitation among women varies widely in SSA, with an overall prevalence of 10.9%—ranging from 50.6% in Liberia to just 0.1% in Senegal. We observed that women with higher levels of acceptance of spousal violence, greater decision-making capacity, and increased general knowledge level were more likely to enter cohabiting relationships. In contrast, women aged 25 and above, those residing

**Data availability statement:** The datasets are publicly available at the DHS repository using the link: http://dhsprogram.com/data/available-datasets.cfm

**Funding:** The author(s) received no specific funding for this work.

**Competing interests:** The authors have declared that no competing interests exist.

**Abbreviations:** aOR: Adjusted Odds Ratios; AIC: Akaike Information Criterion; DHS: Demographic and Health Surveys; SSA: Sub-Saharan African; VIF:variance inflation factor.

in rural areas, those from wealthier households, and those with religious affiliations were less likely to cohabit. Additionally, women whose partners had primary or secondary education had higher odds of cohabitation compared to their counterparts whose partners had no formal education.

## Conclusion

We conclude that women's empowerment plays a significant role in the rising cohabitation rates in SSA. We postulate that addressing adolescent pregnancies could have a significant impact on reducing the practice of cohabitation among women of reproductive age in SSA. Research directions require longitudinal studies to understand the evolving relationship between empowerment and relationship choices, as well as qualitative inquiries to reveal underlying motivations, and comparative analyses across diverse cultural contexts to deepen insights into the interplay between empowerment and cohabitation decisions.

## Background

Globally, the study of marriage has become considerably more intricate in modern times due to unparalleled shifts in the timing, duration, and sequence of intimate co-residential relationships [1,2]. A significant aspect of the evolving landscape of family structures is the growing prevalence of cohabitation [3]. Cohabitation, in this context, refers to the living arrangement of unmarried partners who reside together in a manner resembling that of a married couple, with or without children.

Available evidence indicates that the prevalence of cohabitation has increased significantly in most developed countries [1], however, in the past few years, this increasing prevalence has been observed in sub-Saharan Africa (SSA) [3,4]. For instance, Odimegwu et al. [3] report in their study that 11.8% of marital unions in SSA were cohabitations, with some sub-regional variations where Central Africa reported the highest proportion of cohabitation (21.7%) with West Africa (6.8%) reporting the lowest cohabitation proportions. Similarly, a study conducted in Cameroon [4] revealed that the incidence of cohabitation increased from 15% to 38.9% between 1991 and 2014.

Although cohabitation contradicts the socio-cultural norms and value system surrounding family formation in SSA, its increasing prevalence in the region has been documented to be facilitated by 'secularization, economic constraints and inability to pay bride wealth' [3]. Additionally, the existing body of literature on cohabitation in SSA [3–5] has identified other factors that increase the likelihood of being in a cohabiting union, including older age, early sexual debut, lower educational attainment, unemployment, and belonging to the poorest wealth index. Thus, making cohabitation an important demographic issue of relevance to the wellbeing and welfare of those who contract such unions.

There is a growing interest regarding how economic independence, autonomy in decision-making and other components of women's empowerment influence

cohabitation unions. This probable association between women empowerment and cohabitation dynamics can be viewed from several theoretical perspectives. One of such theories is the rational choice theory (RCT). At its core, RCT assumes that individuals are rational agents who possess preferences that are both stable and ordered [6]. The theory assumes access to relevant information, although some adaptations (e.g., bounded rationality) relax this assumption to accommodate real-world limitations on information processing. In family demography, RCT is particularly valuable for analyzing decisions related to marriage, fertility, divorce, and caregiving [6]. For instance, individuals may weigh the economic, emotional, and social costs of childbearing against perceived benefits, including social status, long-term support, or personal fulfillment. In the context of this study, it can be argued that empowered women are rational actors who are likely to engage in a cost-benefit analysis when choosing between marriage and cohabitation. Traditional marriage, often associated with patriarchal norms and economic dependence, may be perceived as having higher costs (e.g., loss of autonomy, restricted opportunities) and fewer benefits for women with access to education and employment. In contrast, cohabitation may be perceived as offering a more flexible and equitable arrangement that allows women to maintain greater control over their lives while still enjoying the benefits of a partnership. As such, empowerment becomes a tool that places women in a better position evaluate the relative advantages and disadvantages of cohabitation compared to marriage or remaining single.

Beyond the assumptions of RCT, postmodernist theories provide a theoretical underpinning for our study. Central to postmodernist theories is the rejection of a singular "truth" or linear progression of social phenomena [7]. Instead, this theoretical position argues that reality is socially constructed and contingent on historical, cultural, and individual contexts [7]. This means that postmodernism rejects the privilege of the nuclear family as the universal model of intimate relationships. It rather highlights the legitimacy and diversity of alternative arrangements, such as cohabitation, as a reflection of broader societal shifts. Hence, in this view, cohabitation is viewed as a dynamic relationship form that may serve as a precursor to marriage, a substitute for marriage, or a unique partnership model.

Despite the theoretical postulations, a critical gap remains in our understanding of the role of women's empowerment in shaping cohabitation patterns. The existing studies [3–5] in SSA have mainly focused on assessing the trends and determinants of cohabitation. While empowerment has been acknowledged as a critical determinant of various aspects of women's lives including their utilization of maternal healthcare services [8], safe sex negotiations [9], and as a protective factor against intimate partner violence [10], its specific impact on cohabitation dynamics has not been thoroughly explored in this region. This presents a significant knowledge gap that must be filled. We, therefore, examined the association between women empowerment indicators and cohabiting relationships in SSA using data from 13 countries.

## Methods

### Data source and study design

This study is based on data obtained from the most recent standard demographic and health surveys (DHS) conducted between 2018 and 2021 in thirteen (13) sub-Saharan African countries such as Liberia, Benin, Cameroon, Gambia, Guinea, Kenya, Madagascar, Mali, Nigeria, Rwanda, Sierra Leone, Senegal, and Zambia. The DHS of these countries were included in the study because they offer reliable data on marital relations, including other relevant background characteristics, making it appropriate for this study. The surveys involved samples of women within their reproductive age group (15–49 years), selected using a multi-stage stratified cluster sampling procedure to ensure national representation [11]. This involved randomly selecting primary sampling units – mainly clusters – in the first stage and subsequently selecting various households from each of these sampling units in the second stage [10,11]. The sample for the analysis consisted of 124,183 women between the ages of 15 and 49 having information on the outcome of interest, after dropping missing values. Ethical approval for the use of the dataset was not required since it was drawn from a secondary data source. However, permission to use the data was sought from the MEASURE DHS Program. We relied on the Strengthening the Reporting of Observational Studies in Epidemiology (STROBE) guidelines in preparing this paper.

## Study variables and measurements

**Outcome variable.** The outcome variable of interest was cohabitation, which was assessed by determining the count of women within the overall group of women who indicated living together with a partner. In order to gauge the frequency of cohabitation, the study participants were divided into two groups: those who were cohabiting and those who were not. The designation of 'not cohabiting' encompassed all alternative marital statuses, namely widowed, divorced/separated, and never married. Women who were engaged in cohabitation were coded as "1" and "0" otherwise.

**Explanatory variables.** This study primarily focused on women's empowerment, examining it through various measures, aligning with prior research by Yaya et al. [12] and Adde et al. [13], who also considered four empowerment indicators. These indicators encompassed: (1) labor participation, categorizing individuals as either "not working=0" or "employed=1"; (2) acceptance toward spousal violence; (3) decision-making capacity; and (4) general knowledge level. Acceptance toward spousal violence was a composite variable derived from five reasons justifying beating a wife, including going out without permission, neglecting children, arguing with the husband/partner, refusing sexual relations, and burning food, measured as yes = 1 or no = 0. An index was created by summing these responses, resulting in scores from 0 to 5, categorized as "low" (0–1), "medium" (2–3), or "high" (4–5) acceptance of spousal violence. The reliability of this index was checked with a Cronbach's alpha, which shows a value of 0.88, indicating good reliability.

Likewise, general knowledge level was a composite variable derived from factors like education level and media consumption frequency, yielding scores ranging from 0 to 4, categorized as "low" (0), "medium" (1–2), or "high" (3–4) general knowledge level, with a Cronbach's alpha of 0.69, indicating acceptable reliability. Lastly, decision-making capacity assessed authority in healthcare, household earnings, purchases, and family visits, recoded as "yes" (0–1) or "no" (2–4), yielding scores from 0 to 3, categorized as "low" (0), "medium" (1–2), or "high" (3) decision-making capacity, with a Cronbach's alpha of 0.84, indicating good reliability.

Other multiple relevant covariates such as women's age, educational level, type of residence, religion, wealth index, age at first birth, partner's age and level of education were identified and controlled for in our analysis as potential confounders based on previous empirical literature [3].

## Statistical analysis

We conducted a descriptive analysis of the background characteristics of the respondents by cohabitation status. A bar graph was utilized to visually represent the prevalence of cohabitation across the selected countries. A chi-square test was performed to determine statistically significant associations between the outcomes and explanatory variables. Furthermore, multivariate analyses were used separately to examine the association between women empowerment indicators and cohabitation. Additionally, relevant factors were identified and accounted for in the final model. The models were used to calculate unadjusted and adjusted odds ratios with 95% confidence intervals. All estimates were adjusted using sample weighting to correct for any potential bias resulting from under- or over-sampling of participants from the entire population. This was achieved by utilizing the individual sampling weight variable, v005, present in the dataset. Owing to the complex sampling design of the surveys, all analyses were adjusted for clustering at the primary sampling unit level, stratification, and sample weight effects. Before performing the multivariate logistic regression, the possibility of multicollinearity was examined using the variance inflation factor (VIF), which showed a mean score of 5.59, indicating low significant multicollinearity. All analyses were performed using Stata version 17.

## Ethical consideration

We affirm that all procedures were conducted in accordance with the applicable norms and regulations at the time, including obtaining regulatory approval from NHS organizations for research involving human subjects. Every participant provided written consent by signing an informed consent form. The Institutional Review Board of ICF International, along

with the Institutional Review Boards in the respective host countries, have approved the survey protocols. More details regarding DHS data and ethical standards are available at: http://goo.gl/ny8T6X.

## Results

### Descriptive results

Table 1 displays the composition of the women under study, both in terms of weighted and unweighted samples. The table also includes the survey years, spanning from 2017/2018–2021. Among the total sample of 124,183 women, 13,492 were found to be engaged in cohabitation relationships.

Figure 1 represents the prevalence of cohabitation among women in the thirteen (13) countries included in the study. Overall, 10.9% of the women included in the study were found to be in cohabiting relationships. The prevalence of cohabitation ranges from 50.6% in Liberia to 0.1% in Senegal. Equally, more than one-third (35.4%) of the sampled women in Rwanda were engaged in cohabitation relationships. Similarly, about one-fifth of women in Benin and Cameroon were respectively involved in cohabitation relationships.

### Proportional distribution of women involved in cohabitation relationships by women's empowerment indicators and other background characteristics

Table 2 displays the proportion of women involved in cohabitation relationships based on various background characteristics. This includes the chi-square test score along with corresponding p-values, indicating the presence of statistically significant connections between the explanatory variables and the outcome (cohabitation). Apart from the type of place of residence, all chosen explanatory variables exhibited statistically significant associations with the outcome. The analysis revealed that 11.7% of women who were employed were involved in cohabitation relationships compared to 8.8% among women who were not employed. Regarding the perspective on violence, 11.6% of women who held a less accepting attitude towards violence were found to be engaged in cohabitation, whereas this percentage was 8.2% for those with a more accepting attitude. In terms of decision-making capability, 14.4% of the respondents involved themselves in cohabitation relationships, which was notably higher compared to those with limited decision-making capacity (5.5%). A similar pattern emerged when considering the level of knowledge.

**Table 1. Distribution of cohabitation among women in 13 SSA countries.**

| Countries | Survey year | Weighted (N) | Unweighted (N) | Sample of cohabited women (n) |
|---|---|---|---|---|
| Liberia | 2019-2020 | 3,828 | 4,166 | 1,938 |
| Benin | 2017-2018 | 10,111 | 10,016 | 2,127 |
| Cameroon | 2018 | 7,076 | 6,706 | 1,562 |
| Gambia | 2019-2020 | 6,080 | 6,606 | 18 |
| Guinea | 2018 | 6,977 | 7,017 | 144 |
| Kenya | 2022 | 16,867 | 17,266 | 2,093 |
| Madagascar | 2021 | 10,656 | 10,472 | 1,617 |
| Mali | 2018 | 7,701 | 7,341 | 48 |
| Nigeria | 2018 | 26,939 | 26,612 | 860 |
| Rwanda | 2019-2020 | 7,120 | 6,947 | 2,524 |
| Sierra Leone | 2019 | 9,001 | 9,064 | 493 |
| Senegal | 2019 | 4,645 | 4,931 | 6 |
| Zambia | 2018-2019 | 7,180 | 7,039 | 62 |
| **All countries** | | **124,183** | | **13,492** |

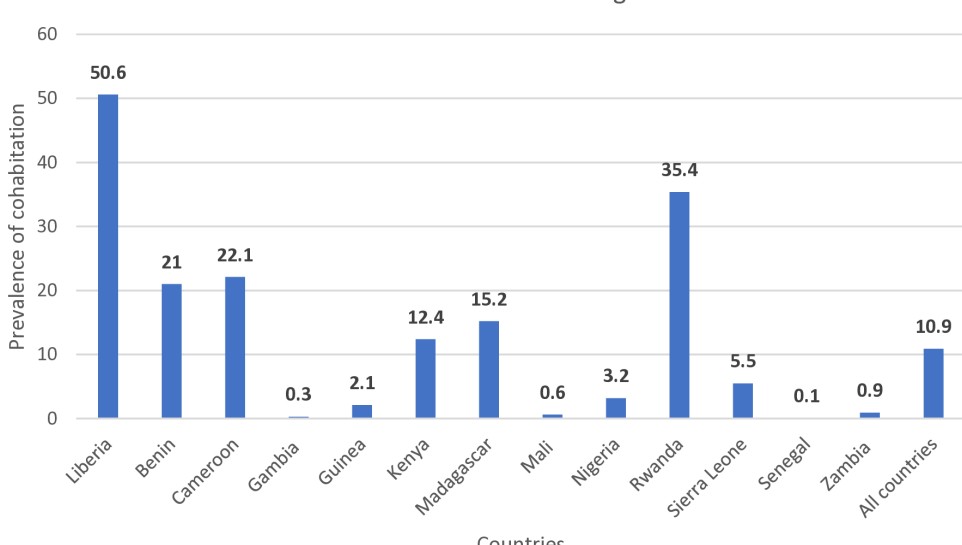

**Fig 1. A bar graph showing the prevalence of cohabitation among women by country of residence in SSA.**

Regarding the covariates, the proportions of cohabiting individuals significantly varied across different age groups. The highest proportion was observed among women aged 15–19 (14.9%), while the lowest proportion was among those aged 25 and above. Notably, cohabitation was less common among Muslim women, with the highest proportion found among women with no religious affiliation (2.5%). The distribution of women in cohabitation relationships was relatively even across different wealth indices.

### Association between women empowerment indicators and cohabitating relationships

The outcomes of the logistic regression analysis investigating the association between women empowerment indicators and cohabitation are presented in Table 3. At this analytical level, two models were employed: Model 1 assesses the connection between women's empowerment and the outcome, while Model II adjusted for additional relevant factors to evaluate this connection. The odds ratios (OR) and their corresponding 95% confidence intervals (CI) were calculated to assess the impact of different variables. In the final model, the analysis shows that women who held medium [aOR = 1.12, CI = 1.05, 1.18] and high levels of acceptance toward spousal violence [aOR = 1.20, CI = 1.12, 1.28], those with medium [aOR = 1.22, CI = 1.14, 1.31] and high [aOR = 1.10, CI = 1.04, 1.18] decision-making capacity, and those with medium [aOR = 1.36, CI = 1.27, 1.46] and high [aOR = 1.32, CI = 1.21, 1.43] levels of general knowledge had higher odds of cohabiting compared to their counterparts with a less acceptance toward spousal violence, limited decision-making capacity, and low levels of general knowledge.

Turning to the variables that were controlled for, higher age was associated with decreased odds of cohabitation. Consequently, when compared to women aged 15–19, those aged 20–24 and 25 years and older were 24% and 56% less likely, respectively, to be involved in cohabitation relationships. The analysis also demonstrated that women living in rural areas had lower odds of cohabitation [aOR = 0.77, CI = 0.73, 0.81] compared to women residing in urban settings. Furthermore, the findings indicated that women who were religiously affiliated were less likely to engage in cohabitation relationships compared to those with no religious affiliation. We found that women from households with the highest wealth index [aOR = 0.67, CI = 0.61, 0.74] had a reduced likelihood of engaging in cohabitation relationships compared to their counterparts from households with the lowest wealth index. Conversely, compared to women who gave birth before the age of 20,

**Table 2. Cohabitation by background characteristics.**

| Explanatory Variables | Proportion of cohabited women | | |
|---|---|---|---|
| | (n) | (%) | P-value |
| **Women empowerment indicators** | | | |
| **Labor force participation** | | | <0.001 |
| *Not employed* | 3,283 | 8.80 | |
| *Employed* | 10,209 | 11.75 | |
| **Acceptance toward spousal violence** | | | <0.001 |
| *Low* | 9,416 | 11.61 | |
| *Medium* | 2,236 | 10.80 | |
| *High* | 1,840 | 8.23 | |
| **Decision-making capacity** | | | <0.001 |
| *Low* | 1,856 | 5.50 | |
| *Medium* | 3,705 | 10.55 | |
| *High* | 7,931 | 14.34 | |
| **General knowledge level** | | | <0.001 |
| *Low* | 1,650 | 7.01 | |
| *Medium* | 7,329 | 11.85 | |
| *High* | 4,513 | 11.63 | |
| **Covariates** | | | |
| **Women's age** | | | <0.001 |
| *15-19* | 744 | 14.86 | |
| *20-24* | 2,557 | 13.96 | |
| *25 and above* | 10,191 | 10.10 | |
| **Type of residence** | | | 0.160 |
| *Urban* | 5,205 | 11.42 | |
| *Rural* | 8,287 | 10.54 | |
| **Religion** | | | <0.001 |
| *No religion[a]* | 918 | 25.01 | |
| *Christians* | 10,716 | 16.88 | |
| *Islam* | 1,360 | 2.49 | |
| *Other religion* | 498 | 20.13 | |
| **Wealth status** | | | <0.001 |
| *Poorest* | 2,515 | 10.05 | |
| *Poorer* | 2,644 | 10.52 | |
| *Middle* | 2,789 | 11.30 | |
| *Richer* | 3,088 | 12.45 | |
| *Richest* | 2,456 | 10.01 | |
| **Age at first birth** | | | <0.001 |
| *Below 20 years* | 7,303 | 10.58 | |
| *20-24* | 4,740 | 11.84 | |
| *25 and above* | 1,450 | 9.56 | |
| **Partners' age** | | | <0.001 |
| *15-19* | 77 | 24.18 | |
| *20-24* | 936 | 23.46 | |
| *25 and above* | 12,479 | 10.41 | |
| **Partners' educational level** | | | <0.001 |
| *No education* | 2,844 | 6.35 | |

*(Continued)*

**Table 2.** (Continued)

| Explanatory Variables | Proportion of cohabited women | | |
|---|---|---|---|
| | (n) | (%) | P-value |
| *Primary* | 4,694 | 15.31 | |
| *Secondary* | 4,746 | 13.55 | |
| *Higher* | 1,207 | 8.80 | |

No religion[a] (Nigeria and Zambia do not have information on this category in the dataset).

#Note: estimates are weighted

those who gave birth between the ages of 20–24 and 25 years and above faced a lower risk of engaging in cohabitation relationships. The findings also highlighted a significant positive link between the educational level of partners and the outcome. Hence, women whose partners had primary and secondary education had higher odds of cohabitation compared to those whose partners had no formal education. Additionally, the findings pointed out disparities in cohabitation across different countries. In comparison to Liberia, women from all the countries studied were less likely to be involved in cohabitation relationships.

## Discussion

This study aimed to examine the association between women's empowerment indicators and cohabitating relationships in SSA. Our findings revealed that 10.9% of women of reproductive age were in cohabitation unions. The observed proportion of cohabitation is similar to what has been reported in a study by [3]. However, the observed proportion of cohabitation among women is lower when compared to other jurisdictions including China (46.4%) [14], Mexico (13.1%) [15], and the UK (23.4%) [16]. The low prevalence of cohabitation in SSA could be explained by the societal non-acceptance of this type of union. Cohabitation in SSA is often considered an unconventional partnership that deviates from the acceptable traditional, religious and cultural norms relating to the formation of marital and/or sexual unions [3,17]. This normative system exacerbates the social stigma of cohabitation which discourages women from cohabiting.

Despite the relatively low prevalence of cohabitation in SSA, the study highlighted significant disparities across countries, ranging from 50.6% in Liberia to 0.1% in Senegal. These contrasting patterns highlight the diverse influence of traditional, religious, and cultural norms across SSA. From a religious perspective, while all religions generally advocate abstinence before marriage, Islam stands out for its strong condemnation of sexual liberalization [18,19]. Cohabitation is explicitly forbidden in Islam, largely due to the pronounced norm of virginity [20], which may explain the low prevalence in predominantly Islamic Senegal. Conversely, Liberia, with its predominantly Christian tradition, may have fewer restrictive norms regarding cohabitation among some Christian denominations. These varying societal norms shape cohabitation patterns, reinforcing the significance of understanding these dynamics to inform culturally sensitive policies and interventions across SSA.

Results from our analyses support the hypothesis that women's empowerment is significantly associated with cohabitation. Theoretically, this aligns with postmodernism perspectives that challenge traditional marriage as the only universal or singular truth [7]. With the exception of labor participation, which was not significant in the adjusted model, higher scores in the remaining three indicators of women empowerment (i.e., acceptance toward spousal violence, decision-making capacity, and general knowledge level) were associated with higher odds of cohabitation. The findings suggest that individuals with high attitudes towards violence are more likely to choose cohabitation as a form of partnership. While there are no clear studies that have investigated this association, our findings corroborate Wong et al.'s [21] study that posits that intimate partner violence is more pervasive in cohabitation unions compared to those who are married. We postulate that women with high attitudes towards violence might have limited conflict resolution skills; as such, cohabitation may be

**Table 3. Bivariate and multivariate logistic regression analysis of the interplay of women empowerment indicators and cohabitating relationship.**

| Variables | Model 1 | | Model II | |
|---|---|---|---|---|
| | **OR** | **95%CI** | **aOR** | **95%CI** |
| **Women empowerment indicators** | | | | |
| **Labor forced participation** | | | | |
| *Not employed* | Ref | Ref | Ref | Ref |
| *Employed* | 1.29*** | [1.24,1.35] | 1.01 | [0.96,1.06] |
| **Acceptance toward spousal violence** | | | | |
| *Low* | Ref | Ref | Ref | Ref |
| *Medium* | 1.02 | [0.97,1.07] | 1.12*** | [1.05,1.18] |
| *High* | 0.79*** | [0.75,0.83] | 1.20*** | [1.12,1.28] |
| **Decision-making capacity** | | | | |
| *Low* | Ref | Ref | Ref | Ref |
| *Medium* | 2.11*** | [1.99,2.24] | 1.22*** | [1.14,1.31] |
| *High* | 2.74*** | [2.59,2.89] | 1.10*** | [1.04,1.18] |
| **General knowledge level** | | | | |
| *Low* | Ref | Ref | Ref | Ref |
| *Medium* | 1.56*** | [1.47,1.64] | 1.36*** | [1.27,1.46] |
| *High* | 1.29*** | [1.21,1.37] | 1.32*** | [1.21,1.43] |
| **Covariates** | | | | |
| **Women's age** | | | | |
| *15-19* | | | Ref | Ref |
| *20-24* | | | 0.76*** | [0.68,0.83] |
| *25 and above* | | | 0.44*** | [0.40,0.48] |
| **Type of residence** | | | | |
| *Urban* | | | Ref | Ref |
| *Rural* | | | 0.77*** | [0.73,0.81] |
| **Religion** | | | | |
| *No religion[a]* | | | Ref | Ref |
| *Christians* | | | 0.54*** | [0.49,0.60] |
| *Islam* | | | 0.14*** | [0.13,0.16] |
| *Other religion* | | | 0.62*** | [0.54,0.74] |
| **Wealth index** | | | | |
| *Poorest* | | | Ref | Ref |
| *Poorer* | | | 1.04 | [0.98,1.11] |
| *Middle* | | | 1.03 | [0.96,1.10] |
| *Richer* | | | 0.98 | [0.91,1.06] |
| *Richest* | | | 0.67*** | [0.61,0.74] |
| **Age at first birth** | | | | |
| *Below 20 years* | | | Ref | Ref |
| *20-24* | | | 0.91*** | [0.87,0.96] |
| *25 and above* | | | 0.74*** | [0.68,0.80] |
| **Partners' educational level** | | | | |
| *No education* | | | Ref | Ref |
| *Primary* | | | 1.09** | [1.03,1.16] |
| *Secondary* | | | 1.36*** | [1.27,1.46] |

*(Continued)*

**Table 3.** (Continued)

| Variables | Model 1 | | Model II | |
|---|---|---|---|---|
| | OR | 95%CI | aOR | 95%CI |
| *Higher* | | | 1.06 | [0.96,1.16] |
| **Country variable** | | | | |
| Liberia | | | Ref | Ref |
| Benin | | | 0.32*** | [0.29,0.35] |
| Cameroon | | | 0.37*** | [0.34,0.41] |
| Gambia | | | 0.00*** | [0.00,0.01] |
| Guinea | | | 0.05*** | [0.04,0.06] |
| Kenya | | | 0.10*** | [0.09,0.11] |
| Madagascar | | | 0.13*** | [0.12,0.15] |
| Mali | | | 0.02*** | [0.01,0.02] |
| Nigeria | | | 0.05*** | [0.05,0.06] |
| Rwanda | | | 0.55*** | [0.50,0.60] |
| Sierra Leone | | | 0.12*** | [0.11,0.14] |
| Senegal | | | 0.00*** | [0.00,0.01] |
| Zambia | | | 0.00*** | [0.00,0.01] |
| ***Constant*** | 0.04*** | [0.04,0.05] | 3.19*** | [2.72,3.74] |
| **Model fitness** | | | | |
| Prob > chi2 | < 0.001 | | < 0.001 | |
| Pseudo $R^2$ | 0.0263 | | 0.2465 | |
| AIC | 81474.19 | | 63110.68 | |

Ref: reference category; AIC: Akaike Information Criterion; OR: odds ratio; aOR: adjusted odds ratio;

***$p < 0.001$,

**$p < 0.010$,

*$p < 0.050$.

perceived as a more flexible and transient arrangement, allowing them to avoid the commitment and challenges of a formal marriage where conflict resolution and compromise might be more critical. More studies are required to fully comprehend the intricate association between attitudes to violence and women's involvement in cohabitation unions.

As indicated, our study suggests that women who had high decision-making capacity and general knowledge levels were more likely to cohabit. Marriage, particularly in SSA, is a family formation structure that is significantly shaped by acceptable sociocultural norms and expectations [22,23]. It means that it is these sociocultural norms that pressure or influence women to get married. However, being empowered to make decisions for oneself, including in matters relating to marriage makes women more assertive to go against long-held social viewpoints about marriage, thereby increasing their likelihood to cohabit. Also, as postulated by the RCT, the empowered women may weigh the economic, emotional, and social costs of childbearing against perceived benefits, including social status, long-term support, or personal fulfillment [6]. This situation is likely to make women who are empowered in decision-making and in terms of their knowledge consider the cost of marriage as high since it has the tendency to be associated with a trade-off of reduced personal independence [3,7]. Thus, explaining the high odds of cohabitation among empowered women.

The study also found significant associations between some covariates and cohabitation. Women aged 25 years and above were less likely to enter cohabitation – a result that is inconsistent with Odimegwu et al.'s study [3] which found higher odds among this age group when compared to those in lower age groups. Perhaps the lower odds of cohabitation

among older women of reproductive age compared to those <20 years could be due to adolescent pregnancy. In most countries in SSA, when adolescent girls get pregnant, they are often forced to move in to cohabit with the male responsible for the pregnancy [24,25]. One study [26] supports this explanation by indicating that the likelihood of cohabitation increased by 61% among adolescents when they were compelled to enter a union due to pregnancy. It is, therefore, not surprising that our study found an inverse association between age at first birth and the likelihood of cohabiting. Having a child at an early age, in SSA, is often considered a deviation from societal values; this may come with some level of stigmatization, shame and ridicule from members of the community [27,28]. It is possible that those who give birth at an early age would enter cohabitation in a bid to somewhat legitimize their pregnancy.

Consistent with previous studies [3,29,30], we found that women with religious affiliation were less likely to engage in cohabitation compared to those who were not religiously affiliated. A possible explanation is that religious teachings often emphasize marriage as a sacred institution. Hence, religious women may hold stronger beliefs in the sanctity of marriage and may view cohabitation as contrary to their religious teachings. This could lead them to choose marriage over cohabitation. Conversely, less religious women may be more open to secularization which is one of the main contributors to the acceptance of cohabitation unions [3,31].

Our study shows that women in rural areas were less likely to cohabit than those in urban areas. Similar findings have been reported in Zambia [23]. Rural areas often tend to have more conservative and traditional values compared to urban areas. Such strong conservative views and traditional values might discourage cohabitation among women and rather promote traditional marriage. In line with a prior study conducted in SSA [3], we found that the odds of cohabitation were significantly low among those in the richest wealth index compared to those in the poorest wealth index. A possible explanation is that people from households with the poorest wealth index may be more likely to cohabit because they cannot afford to get married due to the high cost of the bride price or bride wealth [3,32]. Women who had a partner who had primary or secondary education were more likely to cohabit. The reasons behind this association remain unclear, and further research is needed to enhance our understanding of the link between a partner's level of education and the likelihood of cohabitation

## Strengths and limitations of the study

One of the limitations of this study is the use of a dataset that employs a cross-sectional design. This does not permit us to establish causality between women empowerment and cohabitation. Also, our study does not differentiate between premarital cohabitation and cohabitation after divorce, widowhood or separation. Other cultural values and norms, which may have influenced the outcomes, were not considered due to data limitations. Nevertheless, the sample for this study is large enough to support the extrapolation of the findings to women of reproductive age in SSA. We also employed appropriate analytical tools which ensure the rigor and reliability of the findings.

## Conclusion

The study findings reveal significant variations in the prevalence of cohabitation across the thirteen countries, with the overall rate standing at 10.9%. The prevalence ranged from 50.6% in Liberia to 0.1% in Senegal. The findings demonstrated that women with higher acceptance of spousal violence, greater decision-making capacity, and high general knowledge were more likely to cohabit. Additionally, older women, those living in rural areas, women with religious affiliations, and women from wealthier households had lower odds of cohabitation. Based on the findings, we postulate that women's empowerment contributes significantly to cohabitation union formation in SSA. The findings further connote that addressing adolescent pregnancies would have a significant impact on reducing the practice of cohabitation among women of reproductive age in SSA. We recommend that research directions should include longitudinal studies to understand the evolving relationship between empowerment and relationship choices, qualitative inquiries to reveal underlying motivations, and comparative analyses across diverse cultural contexts to expand insights into the interplay between empowerment and cohabitation decisions.

## Acknowledgments

We are grateful to the DHS Program for providing us with access to the dataset.

## Author contributions

**Conceptualization:** Castro Ayebeng, Joshua Okyere.

**Data curation:** Castro Ayebeng.

**Formal analysis:** Castro Ayebeng.

**Supervision:** Kwamena Sekyi Dickson.

**Writing – original draft:** Castro Ayebeng, Joshua Okyere.

**Writing – review & editing:** Castro Ayebeng, Joshua Okyere, Nancy Arthur, Kwamena Sekyi Dickson.

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
