## [Decision Letter · Decision Letter 0]

17 Dec 2024

PONE-D-24-12068Cohabitation in sub-Saharan Africa: does women empowerment matter? Insights from the demographic and health surveyPLOS ONE

Dear Dr. Ayebeng,

Thank you for submitting your manuscript to PLOS ONE. After careful consideration, we feel that it has merit but does not fully meet PLOS ONE’s publication criteria as it currently stands. Therefore, we invite you to submit a revised version of the manuscript that addresses the points raised during the review process.

**ACADEMIC EDITOR:**Based on the reviewers feedback, I am recommending a minor revision. Please revise and resubmit the manuscript.==============================

We look forward to receiving your revised manuscript.

Kind regards,

Srinivas Goli, Ph.D.

Academic Editor

PLOS ONE

Journal Requirements:

When submitting your revision, we need you to address these additional requirements. 1. Please ensure that your manuscript meets PLOS ONE's style requirements, including those for file naming. The PLOS ONE style templates can be found at https://journals.plos.org/plosone/s/file?id=wjVg/PLOSOne_formatting_sample_main_body.pdf and https://journals.plos.org/plosone/s/file?id=ba62/PLOSOne_formatting_sample_title_authors_affiliations.pdf 2. Please include captions for your Supporting Information files at the end of your manuscript, and update any in-text citations to match accordingly. Please see our Supporting Information guidelines for more information: http://journals.plos.org/plosone/s/supporting-information. 

Additional Editor Comments:

Based on the reviewers feedback, I am recommending a minor revision. Please revise and resubmit the manuscript.

Reviewers' comments:

Reviewer's Responses to Questions

**Comments to the Author**

1. Is the manuscript technically sound, and do the data support the conclusions?

Reviewer #1: Yes

Reviewer #2: Yes

2. Has the statistical analysis been performed appropriately and rigorously? 

Reviewer #1: Yes

Reviewer #2: Yes

3. Have the authors made all data underlying the findings in their manuscript fully available?

Reviewer #1: Yes

Reviewer #2: Yes

4. Is the manuscript presented in an intelligible fashion and written in standard English?

Reviewer #1: Yes

Reviewer #2: Yes

5. Review Comments to the Author

Reviewer #1: The paper highlights the emerging area in family dynamics: cohabitation which is a form of parner union standing as an alternative to marital relationships The author gives a short and concise introduction explaining the increase in cohabitation seen in Sub-Saharan Africa (SSA) countries in conflict with the culture, traditions, and norms. With the support of Becker's economic model, the author establishes the theoretical background and finds evidences for associating women empowerment indicators with cohabitation dynamics in SSA.

1) The primary contradictory finding in the study i.e., high attitude of violence women with more and less likely to have a cohabitation union. This can be more empirically explored or theoretically supported.

2) Coming to the conclusion, the author can add stronger findings with the results in addition to proposing future steps.

3) The introduction part doesnt reflect the major findings of the study.

Reviewer #2: 1. The introduction section is loosely written and needs a robust theoretical background. The introduction is disorganized and lacks sufficient depth, which is critical for formulating research questions in family demography.

2. Established theories on marriage and families in connection with behavioral shifts should be included to strengthen the research framework.

3. The literature review needs to be more organized. A more systematic and comprehensive approach is required to strengthen the paper's foundation.

4. While the paper mentions the economic theory twice, it must cite proper theories or provide theoretical support. A solid theoretical background on marital norms and family formation is essential.

5. The discussion and conclusion sections should interpret the findings in the context of a well-established theoretical framework to enhance the paper's coherence and relevance.

6. The paper needs a country-specific comparison or to express the study comparatively, including the rationale for focusing on the chosen context for the following countries. Otherwise, the significance of the study is unclear.

6. PLOS authors have the option to publish the peer review history of their article (what does this mean? ). If published, this will include your full peer review and any attached files.

**Do you want your identity to be public for this peer review?** For information about this choice, including consent withdrawal, please see our Privacy Policy .

Reviewer #1: No

Reviewer #2: No

---

## [Author Response · Author response to Decision Letter 1]

7 Jan 2025

Reviewer #1:

The paper highlights the emerging area in family dynamics: cohabitation which is a form of partner union standing as an alternative to marital relationships The author gives a short and concise introduction explaining the increase in cohabitation seen in Sub-Saharan Africa (SSA) countries in conflict with the culture, traditions, and norms. With the support of Becker's economic model, the author establishes the theoretical background and finds evidences for associating women empowerment indicators with cohabitation dynamics in SSA.

1) The primary contradictory finding in the study i.e., high attitude of violence women with more and less likely to have a cohabitation union. This can be more empirically explored or theoretically supported.

Response: Thank you for pointing that. We share in your concern. However, we acknowledged that this is counterintuitive and there is not enough empirical evidence or theoretical explanation for it. It is for this reason that we stated: “More studies are required to fully comprehend the intricate association between attitudes to violence and women’s involvement in cohabitation unions”

2) Coming to the conclusion, the author can add stronger findings with the results in addition to proposing future steps.

Response: Thank you. This has been considered in our conclusion. It reads “The study findings reveal significant variations in the prevalence of cohabitation across the thirteen countries, with the overall rate standing at 10.9%. The prevalence ranged from 50.6% in Liberia to 0.1% in Senegal. The findings demonstrated that women with higher acceptance of spousal violence, greater decision-making capacity, and high general knowledge were more likely to cohabit. Additionally, older women, those living in rural areas, women with religious affiliations, and women from wealthier households had lower odds of cohabitation. Based on the findings, we conclude that women empowerment contributes significantly to the practice of cohabitation in SSA. The findings further connote that addressing adolescent pregnancies would have a significant impact on reducing the practice of cohabitation among women of reproductive age in SSA. We recommend that research directions should include longitudinal studies to understand the evolving relationship between empowerment and relationship choices, qualitative inquiries to reveal underlying motivations, and comparative analyses across diverse cultural contexts to expand insights into the interplay between empowerment and cohabitation decisions”.

3) The introduction part doesn't reflect the major findings of the study.

Response: We have now revised the introduction section to reflect the major findings by establishing a theoretical foundation. It reads: “There is a growing interest regarding how economic independence, autonomy in decision-making and other components of women’s empowerment influence cohabitation unions. This probable association between women empowerment and cohabitation dynamics can be viewed from several theoretical perspectives. One of such theories is the rational choice theory (RCT). At its core, RCT assumes that individuals are rational agents who possess preferences that are both stable and ordered [6]. The theory assumes access to relevant information, although some adaptations (e.g., bounded rationality) relax this assumption to accommodate real-world limitations on information processing. In family demography, RCT is particularly valuable for analyzing decisions related to marriage, fertility, divorce, and caregiving [6]. For instance, individuals may weigh the economic, emotional, and social costs of childbearing against perceived benefits, including social status, long-term support, or personal fulfillment. In the context of this study, it can be argued that empowered women are rational actors who are likely to engage in a cost-benefit analysis when choosing between marriage and cohabitation. Traditional marriage, often associated with patriarchal norms and economic dependence, may be perceived as having higher costs (e.g., loss of autonomy, restricted opportunities) and fewer benefits for women with access to education and employment. In contrast, cohabitation may be perceived as offering a more flexible and equitable arrangement that allows women to maintain greater control over their lives while still enjoying the benefits of a partnership. As such, empowerment becomes a tool that places women in a better position evaluate the relative advantages and disadvantages of cohabitation compared to marriage or remaining single.

Beyond the assumptions of RCT, postmodernist theories provide a theoretical underpinning for our study. Central to postmodernist theories is the rejection of a singular "truth" or linear progression of social phenomena [7]. Instead, this theoretical position argues that reality is socially constructed and contingent on historical, cultural, and individual contexts [7]. This means that postmodernism rejects the privileging of the nuclear family as the universal model of intimate relationships. It rather highlights the legitimacy and diversity of alternative arrangements, such as cohabitation, as a reflection of broader societal shifts. Hence, in this view, cohabitation is viewed as a dynamic relationship form that may serve as a precursor to marriage, a substitute for marriage, or a unique partnership model in its own right.”

Reviewer #2:

1. The introduction section is loosely written and needs a robust theoretical background. The introduction is disorganized and lacks sufficient depth, which is critical for formulating research questions in family demography.

Response: We have now revised the introduction section to reflect this. It reads: “There is a growing interest regarding how economic independence, autonomy in decision-making and other components of women’s empowerment influence cohabitation unions. This probable association between women empowerment and cohabitation dynamics can be viewed from several theoretical perspectives. One of such theories is the rational choice theory (RCT). At its core, RCT assumes that individuals are rational agents who possess preferences that are both stable and ordered [6]. The theory assumes access to relevant information, although some adaptations (e.g., bounded rationality) relax this assumption to accommodate real-world limitations on information processing. In family demography, RCT is particularly valuable for analyzing decisions related to marriage, fertility, divorce, and caregiving [6]. For instance, individuals may weigh the economic, emotional, and social costs of childbearing against perceived benefits, including social status, long-term support, or personal fulfillment. In the context of this study, it can be argued that empowered women are rational actors who are likely to engage in a cost-benefit analysis when choosing between marriage and cohabitation. Traditional marriage, often associated with patriarchal norms and economic dependence, may be perceived as having higher costs (e.g., loss of autonomy, restricted opportunities) and fewer benefits for women with access to education and employment. In contrast, cohabitation may be perceived as offering a more flexible and equitable arrangement that allows women to maintain greater control over their lives while still enjoying the benefits of a partnership. As such, empowerment becomes a tool that places women in a better position evaluate the relative advantages and disadvantages of cohabitation compared to marriage or remaining single.

Beyond the assumptions of RCT, postmodernist theories provide a theoretical underpinning for our study. Central to postmodernist theories is the rejection of a singular "truth" or linear progression of social phenomena [7]. Instead, this theoretical position argues that reality is socially constructed and contingent on historical, cultural, and individual contexts [7]. This means that postmodernism rejects the privileging of the nuclear family as the universal model of intimate relationships. It rather highlights the legitimacy and diversity of alternative arrangements, such as cohabitation, as a reflection of broader societal shifts. Hence, in this view, cohabitation is viewed as a dynamic relationship form that may serve as a precursor to marriage, a substitute for marriage, or a unique partnership model in its own right.”

2. Established theories on marriage and families in connection with behavioral shifts should be included to strengthen the research framework.

Response: We have now positioned our study on postmodernism and the rationale choice theory.

3. The literature review needs to be more organized. A more systematic and comprehensive approach is required to strengthen the paper's foundation.

Response: Thank you for the comment. The literature reviewed for establishing the purpose of the study has been properly organized.

4. While the paper mentions the economic theory twice, it must cite proper theories or provide theoretical support. A solid theoretical background on marital norms and family formation is essential.

Response: As earlier indicated, we now position the study postmodernism and the rationale choice theory which are key in explaining marital and family dynamics.

5. The discussion and conclusion sections should interpret the findings in the context of a well-established theoretical framework to enhance the paper's coherence and relevance.

Response: We have ensured that the discussion is rooted in well-established theories.

6. The paper needs a country-specific comparison or to express the study comparatively, including the rationale for focusing on the chosen context for the following countries. Otherwise, the significance of the study is unclear.

Response: Thank you for this critical insight. We have addressed this comment which reads “Despite the relatively low prevalence of cohabitation in SSA, the study highlighted significant disparities across countries, ranging from 50.6% in Liberia to 0.1% in Senegal. These contrasting patterns highlight the diverse influence of traditional, religious, and cultural norms across SSA. From a religious perspective, while all religions generally advocate abstinence before marriage, Islam stands out for its strong condemnation of sexual liberalization [18,19]. Cohabitation is explicitly forbidden in Islam, largely due to the pronounced norm of virginity [20], which may explain the low prevalence in predominantly Islamic Senegal. Conversely, Liberia, with its predominantly Christian tradition, may have fewer restrictive norms regarding cohabitation among some Christian denominations. These varying societal norms shape cohabitation patterns, reinforcing the significance of understanding these dynamics to inform culturally sensitive policies and interventions across SSA”.

---

## [Editor Report · Decision Letter 1]

18 Mar 2025

Cohabitation in sub-Saharan Africa: does women empowerment matter? Insights from the demographic and health survey

PONE-D-24-12068R1

Dear Dr. Ayebeng,

We’re pleased to inform you that your manuscript has been judged scientifically suitable for publication and will be formally accepted for publication once it meets all outstanding technical requirements.

Kind regards,

Srinivas Goli, Ph.D.

Academic Editor

PLOS ONE
---

## [Editor Report · Acceptance letter]

PONE-D-24-12068R1

PLOS ONE

Dear Dr. Ayebeng,

I'm pleased to inform you that your manuscript has been deemed suitable for publication in PLOS ONE. Congratulations! Your manuscript is now being handed over to our production team.

Kind regards,

on behalf of

Dr. Srinivas Goli

Academic Editor

PLOS ONE